# Nociception in Chicken Embryos, Part II: Embryonal Development of Electroencephalic Neuronal Activity *In Ovo* as a Prerequisite for Nociception

**DOI:** 10.3390/ani13182839

**Published:** 2023-09-07

**Authors:** Sandra Kollmansperger, Malte Anders, Julia Werner, Anna M. Saller, Larissa Weiss, Stephanie C. Süß, Judith Reiser, Gerhard Schneider, Benjamin Schusser, Christine Baumgartner, Thomas Fenzl

**Affiliations:** 1Department of Anaesthesiology and Intensive Care, School of Medicine, Technical University Munich, 81675 Munich, Germany; s.kollmansperger@outlook.de (S.K.); malteanders@gmail.com (M.A.); gerhard.schneider@tum.de (G.S.); 2Clinical Development and Human Pain Models, Fraunhofer Institute for Translational Medicine and Pharmacology ITMP, 60596 Frankfurt, Germany; 3Center for Preclinical Research, Technical University of Munich, 81675 Munich, Germany; julia.werner@tum.de (J.W.); anna.saller@tum.de (A.M.S.); larissa.weiss@tum.de (L.W.); stephanie.suess@tum.de (S.C.S.); judith.reiser@tum.de (J.R.); christine.baumgartner@tum.de (C.B.); 4Department of Molecular Life Sciences, Reproductive Biotechnology, School of Life Sciences Weihenstephan, Technical University Munich, 85354 Freising, Germany; benjamin.schusser@tum.de

**Keywords:** EEG, nociception, pain, embryo, development, *Gallus gallus domesticus*

## Abstract

**Simple Summary:**

Even today, we do not know from which point the chicken embryo is able to process and feel pain. This is of special interest as worldwide, millions of male embryos are killed before hatching. This work aimed to examine when during the development of the embryo the brain shows normal activity, based on EEG recordings. The data strongly suggest developmental day 13 as the earliest embryonal stage being able to process pain. These results may support legislative processes establishing updated laws on animal welfare.

**Abstract:**

Chicken culling has been forbidden in Germany since 2022; male/female selection and male elimination must be brought to an embryonic status prior to the onset of nociception. The present study evaluated the ontogenetic point at which noxious stimuli could potentially be perceived/processed in the brain *in ovo*. EEG recordings from randomized hyperpallial brain sites were recorded *in ovo* and noxious stimuli were applied. Temporal and spectral analyses of the EEG were performed. The onset of physiological neuronal signals could be determined at developmental day 13. ERP/ERSP/ITC analysis did not reveal phase-locked nociceptive responses. Although no central nociceptive responses were documented, adequate EEG responses to noxious stimuli from other brain areas cannot be excluded. The extreme stress impact on the embryo during the recording may overwrite the perception of noniceptive stimuli. The results suggest developmental day 13 as the earliest embryonal stage being able to receive and process nociceptive stimuli.

## 1. Introduction

Chicken culling has been forbidden in Germany since January 2022. Until this date, around 45 million male birds were killed every year directly after hatching, as raising male layer-type chickens is not profitable for the industry [1]. In recent years, our society gained a clear understanding that sex selection must be brought forward to the embryo status, leaving hatched birds untouched. During *in ovo* sex determination, the sex can be identified early before hatching for example using endocrinological or spectroscopic procedures, so that the incubation of eggs containing male chickens can be prevented [2] as early as possible. Animal welfare and our ethical conscience admonish our society to make sure that sex selection and reasonable killing at this developmental stage must exclude nociceptive perception.

Several studies have indicated that birds can perceive pain in the same way as mammals [3]. This is not limited to behavioural and physiological responses to various nociceptive stimuli, which elicit similar responses as observed in mammals [4,5]. More importantly, it also includes the fact that in birds, cutaneous mechanical, thermal, chemical and polymodal nociceptors have been identified [4,6,7]. As pain includes a subjective component, it is virtually impossible to quantify pain perception deriving from the neuronal activity of nociceptors, especially without communicating through speech. To overcome this limitation, pain research distinguishes between automatic, unconscious recognition of stimuli and thereby induced transmissions of neuronal signals (nociception) and conscious perception of pain. While nociception is the peripheral recognition of potentially tissue-damaging (noxious) stimuli by nociceptors and their transmission through the nociceptive nervous system towards the central nervous system [8], pain is characterized by a subjective, conscious and central sensation, usually triggered by nociception. Pain only arises through the subjective, conscious perception of nociception, which requires a functional brain and its centralized interpretation of nociception [9,10]. In animal research, the most reliable and accepted method to record nociceptive stimuli together with their adequate neuronal answer is the electroencephalogram [6,11,12,13]. 

The development of the chicken embryo and its nervous system is a gradual process, e.g., from the fifth day of incubation, spontaneous movements of the embryo are possible [14]. However, as the nervous system of the chicken embryo is still less developed at this time of embryogenesis, nociception is highly unlikely [15,16]. Previous studies investigating the onset of the first spontaneous EEG activity were inconsistent in their results; thus, developmental day 11 [17], day 12 [18] and day 13 [19] have been identified as the EEG onset. 

Summarized, due to a lack of consistent data on the development of the neuronal system of chicken embryos including peripheral receptors together with sensory/motor pathways and central processing, the currently very limited knowledge does not allow a precise statement on the physiological onset of nociception. This is even more true for the potential capability of pain perception at the central level. The main focus of the present study was to determine the onset of the EEG signal in chicken embryos as a physiological prerequisite for nociception. The second goal was to evaluate whether standardized painful stimuli may trigger subtle changes in epidural EEG signals.

## 2. Materials and Methods

### 2.1. Animals

A total of 361 Lohman Selected Leghorn chicken embryos (TUM Animal ResearchCenter, Versuchsstation Thalhausen, Technical University of Munich) between developmental days 7–19 (ED7–ED19) were used in the experiments (see Table 1 for details). Sexing was performed macroscopically in 280 embryos (♂ = 144 (51.43%), ♀ = 136 (48.58%)) between ED12–ED19. 

The fertilized eggs were disinfected (Röhnfried Desinfektion Pro), labeled and stored at 15 °C (embryogenesis put on hold) for further treatment. Within a week, the experimental animals were moved to an incubator (Favorit Olymp 192 Spezial, HEKA—Brutgeräte, Rietberg, Germany, temperature 37.8 °C, air humidity 55%) and assigned to the developmental day ED0. After three days the eggs were windowed, treated with 0.5 mL Penicillin-Streptomycin (10,000 units penicillin, 10 mg streptomycin/mL, P4333—100 mL Sigma-Aldrich, Darmstadt, Germany) and kept in the incubator until starting the first EEG recordings at day ED07. After termination of the experiments, a lethal anaesthesia was applied via intravenous injection of Pentobarbital-Sodium (Narcoren: ED07–ED19: 16 g/100 mL in 0.1–0.2 mL), followed by decapitation. Developmentally critical stages (ED12, ED13, ED19, see results for details) were additionally defined by a more detailed staging method [20].

### 2.2. EEG Hardware and EEG Recordings

During experiments the chicken embryos were transferred from the incubator to the experimental setup and kept at a mean temperature of 37.5 °C (±2 °C) and a room humidity of 42%. The embryos were given 5 min before preparation to adapt to the environmental changes in light and humidity. The head of the embryo was gently grabbed and brought to the edge of the egg and then fixed for EEG recordings in a way that kept the head and body on the same horizontal level to minimize additional strain on the vascular system [21,22]. 

After fixation of the head, the EEG electrodes were placed epidurally at the cerebellum (electrode EEG1), rostrally at multiple hyperpallial sites (electrode EEG2) and above the optic lobe (reference electrode REF). Additional recordings were performed with EEG1 on the optic lobe [18]. Custom-made gold electrodes were applied, for details refer to [23,24,25,26,27,28]. Each EEG recording consisted of 2 min of basal EEG recordings, followed by 8 min stimulation and another 2 min of basal EEG (Figure 1). The raw EEG data were processed through a pre-amplifier (custom-made, amplification: 1×, npi electronics, Tamm, Germany), amplified (DPA-2FL, npi electronics, Tamm, Germany) with an amplification rate of 1000× (hardware bandpass filter: 0.1 Hz–100 Hz, notch filter @50 Hz) and digitalized @500 Hz (Power1401, CED, Cambridge Electronic Design Limited, Milton/Cambridge, UK) for offline analysis. The recording software (Spike2, CED, Cambridge Electronic Design Limited, Milton/Cambridge, UK) was TTL-synchronized with the stimulation hardware. 

Room temperature and room humidity, egg temperature and moisture of the embryonal brain surface was monitored very closely, as dehydration of the brain surface and electrodes may lead to artifacts and changes in electrical activity [19,29].

### 2.3. Standardized Thermal and Electrical Stimulation

For thermal stimulation, 40 subsequent thermal stimuli were applied with an inter-stimulus interval of 10 s, using a Peltier-element-based thermal stimulation device (TCS, QST.Lab, Strasbourg, France). The stimulation device had 0.25 cm^2^ at the tip of the probe, a heating speed of 41 °C/s (see Figure 1) and a final temperature of 51 °C. Peak temperature during contact heat stimulation was chosen from the literature for nociceptive thermal thresholds [6,30,31]. For electrical stimulation, an electrically isolated constant current stimulator (ISO-STIM 01B, npi electronics, Tamm, Germany), with a constant stimulation current of 1 mA [32,33] (pulse duration: 150 µs, inter-pulse interval: 5 ms, pulse train duration: 40 ms, inter-stimulus interval: 5 s) was used. The composition of the pulse train derived from previous studies applying electrical microstimulations [34,35,36]. 

### 2.4. EEG Data Processing

Raw EEG recordings below 25 µV during a minimum of 90% individual recording time and EEG recordings with amplitudes exceeding 500 µV were rejected. 

Selected EEG text files were transferred into a vector file (MatLab, MathWorks, Natick, MA, USA), and subsequently imported to the MATLAB toolbox EEGLAB [37] for analysis. As most automated artifact rejection routines such as artifact subspace reconstruction (ASR) are only validated for human data [38], datasets that exceeded ±500 µV in amplitude for more than 10% of the recording time were manually rejected for analysis. From all manually selected datasets, the EEG signal from −1 s to +2 s around the onset of each stimulus was epoched. Event-related spectral perturbation (ERSP) and inter-trial coherence (ITC) were calculated [39,40] using EEGLAB’s *newtimef*-function. A divisive baseline from −1 s to 0 s, a resolution in time of 400 points from −1 s to +2 s and a frequency resolution of 200 points between the frequencies of 3 Hz and 100 Hz [41,42] were chosen. EEG signals from EEG1 were analyzed with a wavelet transform portion of the *newtimef*-function with 3 cycles at the lowest frequency of 3 Hz and 20 cycles at the highest frequency of 100 Hz. In the ERSP results, any deactivation or activation below a threshold of −2 dB or above a threshold of +2 dB was considered as a response to the stimulus [39,40,43]. For the analysis of all baseline EEG recordings and stimuli-locked EEGs, 6 randomly selected datasets from each development stage (ED07–ED19) were selected for further analysis. Spectral EEG parameter were analyzed as power spectral density (PSD) with the *pwelch* function from the MATLAB Signal Processing Toolbox and plotted as density spectral arrays (DSA) in a logarithmic (log10) average across all embryos of a particular developmental day.

### 2.5. Physiological Anticipations

The median ERSP and ITC data are only shown for d19 embryos, as any EEG response on external stimuli was anticipated at the latest development stage, shortly before hatching. The phase-locked response in the EEG after electrical stimulation was expected rather immediately after the onset of the stimulus (below 100 ms). The EEG response after thermal stimulation was expected well after the onset of the stimuli due to ∆T_heating_ of the Peltier-element (between 100 ms and 800 ms) [39,40,43].

### 2.6. Statistics

Only the minimum/maximum ERSP and ITC values and their respective 25% and 75% quartiles, as well as the time and frequency at which they occurred, are presented, as this does not depend on the chosen window size when extracting ERSP data. For the evaluation of differences in the spectral power features, calculation of the area under the curve (AUC) of the receiver-operating characteristic (ROC) for each bin with a frequency resolution [(125/512) Hz] of PSD and 10 k-fold bootstrapped 95% confidence intervals (CI) were performed using the MATLAB-based MES toolbox. A difference between the two distributions was considered significant if the 95% CI did not contain levels above 0.5. The significance level was set to *p* < 0.05. For the statistical comparison of the PSD averages from the small sample size (n = 6), the non-parametrical Mann–Whitney *U* test [44] with its suitability for the analysis of EEG data [45] was applied. Some relevant data may have been missed but the influence of the testing procedure and its statistical results did not have any influence on the general EEG findings and spectral analyses.

### 2.7. Histological Procedures

Following termination of the EEG recordings, brains from ED7, ED12, ED13 and ED19 were removed from the skull and transferred to paraformaldehyde (4% PFA @1x PBS, Sigma-Aldrich) for at least 24 h. After transferring the brains to sucrose solution (30%), the brains were kept at 4 °C. Before slicing (cryotome @100 µm), the brains were mounted in gelatine (60 g gelatine, 50 g sucrose, 0.25 mL Triton X100, 500 mL mQ H_2_0) and transferred again to the PFA and sucrose bath. For anatomical analysis, a standard Nissl-staining protocol (cresyl violet staining) was applied to the anatomical slices.

## 3. Results

### 3.1. Basal EEG Activity

In Figure 2, representative 15 s sections of raw EEG data from three randomly selected datasets (3 × 13 embryos) are shown for D7–D19. The onset of prominent EEG activity can be clearly attributed to ED13. 

Table 2 lists the median and the percentiles (25%; 75%) of the delta band (1–4 Hz) for each development stage from all 6 embryos included in Figure 3. Between ED12 and ED13, the average power of the delta band increased by more than 20 dB in absolute terms (*p =* 0.0022). An additional significant increase of approximately 6 dB (*p* = 0.0043) in the delta band was found between ED10 and ED11.

### 3.2. Electrical and Thermal Stimulation

Figure 4 represents the median event-related spectral perturbation and inter-trial coherence for thermal and electrical stimulation. After applying a threshold of ±2 dB to unmask stimulus-related EEG activities, a single local maximum of 2.79 dB [−2.15 dB/5.16 dB] at 6.41 Hz and 1052 ms was measured after thermal stimulation. After electrical stimulation, a local maximum of 2.23 dB [0.68 dB/2.23 dB] at 13.24 Hz and 543 ms was detected. No further ERSP responses were found.

A local ITC maximum for an thermal stimulation of 0.38 [0.27/0.42] was detected at 6.41 Hz and 797 ms; the local ITC maximum for an electrical stimulation of 0.23 [0.11/0.29] was present at 28.35 Hz and 0.33 Hz. Both local ITC maxima did not correspond to an ERSP response, i.e., a deactivation or activation that exceeds −2 dB or 2 dB, respectively. ITC analysis revealed a low degree of phase locking in the analyzed range of time and frequency. The barely visible local ITC maximum deriving from thermal stimulation indicates that the phase of the oscillation following our stimulus is not completely random. 

### 3.3. Histological Verification

The onset of meaningful EEG activity seems to correspond roughly with the histological data. At ED13, the embryonal development of central neuronal structures is well advanced, anticipating the expression of all neuronal structures within the embryonal brain at ED19 (see Figure 5).

## 4. Discussion

The present study evaluated the neuronal development of an embryonal chicken brain at the level of the EEG. A relatively clear onset of a physiologically meaningful EEG activity could be attributed to ED13. The manifestation of this neuronal activity was shown in the present study until ED19. Electrical and thermal stimuli did not elicit any notable temporal and spectral changes in the corresponding EEGs. 

### 4.1. Basal EEG

The onset of physiologically relevant brain activity in the present study could be reliably demonstrated in various anatomical areas of the hyperpallium from ED13 onwards. Compared to raw EEG signals recorded 2 days after hatching [47], the EEG amplitudes in the embryo show similar temporal and spectral features. In 2-dayold chickens as well as in embryonal stages ED13–ED19, the EEG reach amplitudes of ±100 µV to ±200 µV, although the highest amplitudes were present prior to hatching and not during the first few days after hatching. Interestingly, such decreasing EEG amplitudes were also documented between 2-day-old chickens and 8-week-old chickens with an average amplitude below ±100 µV [47]. Averaged frequency spectra from 2-day-old chickens show a low power maximum around 5–10 Hz with decreasing power towards 40 Hz [47] at frontal recording sites. The embryonal spectral maxima at ED19 were slightly lower within a range from 0.1 Hz to 6 Hz with a maximum at the delta band. A shift from the embryonal delta band towards a dominant but blurry theta/alpha band immediately after hatching may be due to the potential role of the alpha band to act as an attentional suppression mechanism during the selection or elimination of objects or features during cognitive tasks [48]. Whether the embryonal dominant delta band resembles sleep-likes states such as slow-wave sleep [24,49,50,51] remains a functional enigma. Early findings from embryonal EEG recordings demonstrated spontaneous neuronal activity between ED13 and ED16 [52]. In contrast, the dominant frequency band at developmental day 15 was around 4 Hz to 7 Hz [52], shifting towards higher frequencies close to hatching, which is in line with our findings and may resemble post-embryonal findings from others [47]. Why we found the earliest spontaneous EEG activity already around ED13, whereas Peters and co-workers did not report any electrical discharges of the cerebral lobes before day 14, is not clear. One reason could be that we applied EEG recordings continuously on every developmental day from ED7 to ED19. Peters and co-workers only reported data from day 6, day 8, day 10, day 13, day 16 and the first post-embryonal day. Important but minor developments may have been missed. In our studies, we used consecutive numbering starting with ED0 at the first embryonal breeding day. Whether this numbering and the corresponding staging was applied, or numbering began at ED1 was not documented in other studies [52]. Another reason for these differences may be due to the breeding lines used in different experiments and its potential subtle temporal aberrancy in their embryogenesis of the brain. For example, it is known from adult mice that temporal and spectral features of the EEG differ significantly between closely related breeding lines (Huber, 2000 #6179). It is conceivable that neuronal embryogenesis may also differ between different chicken breeding lines *in ovo*.

Apart from such differences, the global features of the late embryonal EEG are similar to basal EEGs derived from adult chickens. Interestingly, amplitudes above ±200 µv as recorded from our ED17–ED19 embryos were reported in resting adult chickens (Ookawa T., 1965 #8834). At this behavioral state, the dominant frequencies in adult chickens were 3–4 Hz and 6 Hz–12 Hz, respectively. Similar data were reported from newly hatched chickens [19]. Nevertheless, even at ED19, the individual EEGs were highly variable, which is consistent with earlier publications and observations in adolescent chickens [17,21,53,54].

The neuronal development, as expressed in the global embryonal EEG activity reported in the present study, seems to correlate with the development of various brain structures. From developmental day 8 onwards, a mass migration of neuroblasts takes place, which is completed around day 11 with segregation along the dorsolateral walls of the cerebrum [55]. By developmental day 12, the diencephalon has undergone a complete differentiation of nuclei [52,56], potentially setting the stage for physiological neuronal activities as presented for ED13. 

From ED07 to ED12, a consistent signal at 16⅔ Hz was frequently recorded, which was covered by more dominant domains from ED13 onwards. Although the literature is very sparse [57] and partly not-peer-reviewed [58], one external source for this very particular frequency recorded may have been subway tracks run by 15 kV AC at 16⅔ Hz in close vicinity to our laboratory. 

### 4.2. Electrical and Thermal Stimulation

Although the EEG maturates during the last week *in ovo*, variations in the electrical patterns are not always correlated with spontaneous motor activity [59] and motility patterns persisted unchanged in total absence of the cerebral EEG [21], raising the principal question in how far an embryonal EEG also does not mirror peripheral sensory input. The situation is much clearer in the adult bird. Physiological responses including spectral changes in the EEG to nociceptive stimuli have been described for awake birds [4,5,60], which are consistent with those observed in awake mammals [13,39,40,43,60]. The neuroanatomical prerequisites such as cutaneous mechanical, thermal, chemical and polymodal nociceptors are present and respond to external stimulation similar to mammalian nociceptors [3,60,61]. 

Stimuli-related electrical potentials in the mammalian brain are typically found in the somatosensory, insular, cingulate, frontal and parietal cortical network [62]. The avian hyperpallium, nidopallium and mesopallium have been proposed to be homologous to the mammalian somatosensory cortex [60,63,64,65]. 

In mammals and birds, somatosensory information is processed from the deeper layers of the thalamus [63,65] with its various nuclei based on their location, pattern of sensory inputs and its embryological derivation [66,67,68] towards the pallium (hyperpallium in birds). The hyperpallium apicale (and the caudomedial nidopallium) seem to be promising areas for avian stimulus and nociceptive processing [64,69,70,71], assumingly representing mammalian sensomotoric functionality. 

Even so, the hyperpallial and nidopallial recording sites seem appropriate, no global EEG responses to the noxious stimuli could be recorded from others [30,31] and in the present study. The requirements to record *in ovo* EEG signal may create their own limitations. Assuming that the embryo is capable of processing nociceptive sensations and stimuli from ED13, the preparation of the embryo itself to record a clear EEG may drive the perceptive capacity already to its limits. Experimentally applied nociceptive stimuli may then have a perceptive threshold below the impact of the preparation of the embryo itself. These nociceptive stimuli could as well be above a given threshold, but the EEG may already be enhanced with nociceptive sensations from the preparation. Either way, subtle changes in the EEG according to an experimental stimulus may be uncovered. At present, no stereotaxic atlas of the chicken embryo is available to record along the spinothalamic tract and its thalamic and striatal projections to overcome this dilemma.

### 4.3. Conscious Pain Perception

Assuming that changes in the cortical activity due to nociceptive stimulation are based on the cognitive perception of pain [60,72], experiments applying a minimal anesthesia protocol were performed in several species [60,73,74,75,76,77]. To our knowledge, only one study used this anesthetic protocol in birds and found no consistent evidence of nociception after thermal, electrical, or mechanical stimulation [60]. These results demonstrate either the absence of nociceptive-driven spectral changes in birds or, more likely, a conscious perception of noxious stimuli [60]. This may raise the question of how far embryos and fetuses possess consciousness. The present literature widely spreads from consciousness being only present immediately after birth [78] across the morality of embryo usage in research [79] towards the general and unsolved question what consciousness really means [80,81,82,83,84]. This question by far goes beyond the scope of the present study, especially when we ask about potential consciousness *in ovo*. 

### 4.4. Selection of the Embryonal Timeframe for EEG Recordings

In birds, C and Aδ axons along the spinothalamic tract terminate at peripheral nociceptive receptors connecting the peripheral nervous system with central regions of the avian brain [85]. These afferent fibers start developing around day 4 in the embryo, including functional multisynaptic reflex arcs and sensomotoric coupling *in ovo* around day 7 [16,78,86], excluding EEG recording before this stage. This neuroanatomical gestation is in line with the selective start of EEG recordings at ED7.

### 4.5. Histological Verification

Assuming that at developmental day 19 the neuronal prerequisites to detect, transmit and process nociceptive stimuli are fully established, a simple anatomical comparison as shown in Figure 5 indicates a general ability for nociception already around day 13. All major structures of the hyperpallium [15] are clearly visible, suggesting also a physiologically similar EEG at day 13 and day 19. The anatomical part of the study was not designed to focus on the morphological development of the brain, but rather being anatomically supportive for a functional EEG. The anatomical development per se would be very interesting, but this would have been out of the focus of a functional EEG study. Further acute slice recording and neuroanatomical verification is needed in the future to support this assumption.

## 5. Conclusions

The present work suggests the onset of a meaningful EEG at the developmental ED13 in the chicken embryo. Is this an adequate indicator for the processing of nociceptive stimuli or even the perception of pain? The literature suggests a central processing of nociceptive information to establish the sensation of acute pain. Based on the present data, this seems unlikely to be before ED13. A direct EEG-based documentation of central nociceptive processing or the perception of pain was not possible in the chicken embryo *in ovo*. To overcome this limitation, we suggest establishing in vivo recordings of neuronal activity upon nociceptive stimuli starting at the level of the peripheral receptors, proceed along the ascending projections towards the developing central nervous system. The establishment of a stereotactic embryonal atlas and acute slice electrophysiology along the embryogenesis together with the present findings have the potential to overcome this limitation. 

## Figures and Tables

**Figure 1 animals-13-02839-f001:**
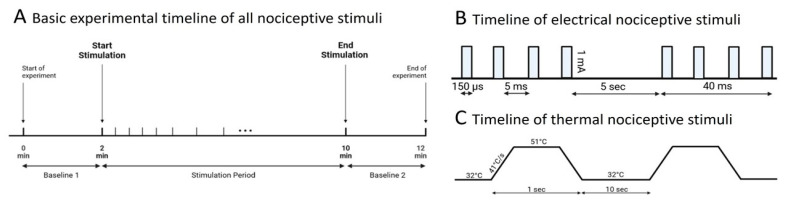
Experimental timeline for EEG recordings in chicken embryos. (**A**) The total duration of a single recording was 12 min, starting and ending with 2 min of baseline EEG recordings. The stimulation duration was 8 min. (**B**) Electrical stimulation was administered at 1 mA, pulse duration: 150 µs, inter-pulse interval: 5 ms, pulse train: 40 ms at 5 s, 90 repetitions/stimulus. (**C**) Thermal stimuli were given at 51 °C with a heating rate of 41 °C/s for 1 s and repeated every 10 s for 40 times. Basal temperature was kept at 32 °C.

**Figure 2 animals-13-02839-f002:**
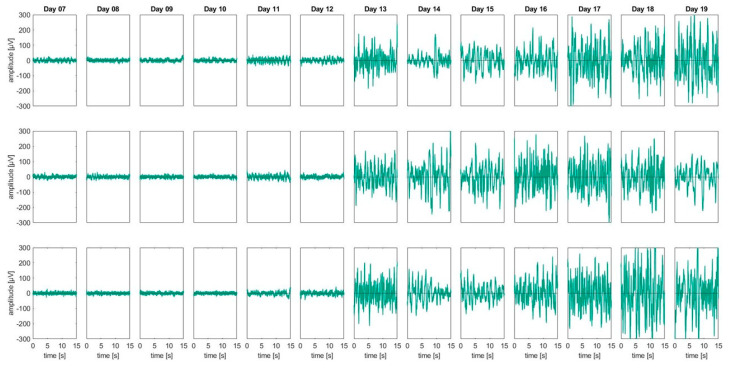
Raw EEG data: An overview of 15 s of raw EEG from three randomly chosen embryo datasets at development stages ED07-ED19. An onset of physiological EEG signatures is prominently visible from ED13 and onwards. The raw EEG amplitudes from ED07 until ED12 partly exceeded ±50 µV, but never exceeded ±100 µV, randomly fluctuating around baseline (0 µV). A strong increase in the EEG signal can be seen from ED13-ED19, with an amplitude regularly exceeding ±200 µV. The plots do not represent longitudinal recordings from ED07-ED19 within one embryo. For each day an individual embryo was recorded and added to a longitudinal graphical presentation representing the global findings.

**Figure 3 animals-13-02839-f003:**
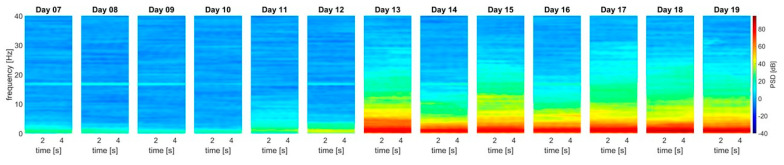
**Top.** Spectral power density: ED07-ED12 revealed no prominent power in all relevant frequency bands, apart from some minor but consistent oscillations in the low delta regions around 1–2 Hz and an isolated signal at 16.33 Hz. The onset of slow delta oscillations is visible from ED13 onwards. For each developmental day, data from 6 representative EEGs were processed. **Bottom**: Boxplots illustrating the median (red line), the 25% and 75% percentiles (lower and upper box end) and the minimum/maximum values (lower and upper whisker) for the delta band power. Red crosses indicate outliers. AUCED10/ED11: 0.97 [0.83, 1], AUCED12/ED13: 1 [1, 1], only significant AUCs reported (refer to Table 2 for other data).

**Figure 4 animals-13-02839-f004:**
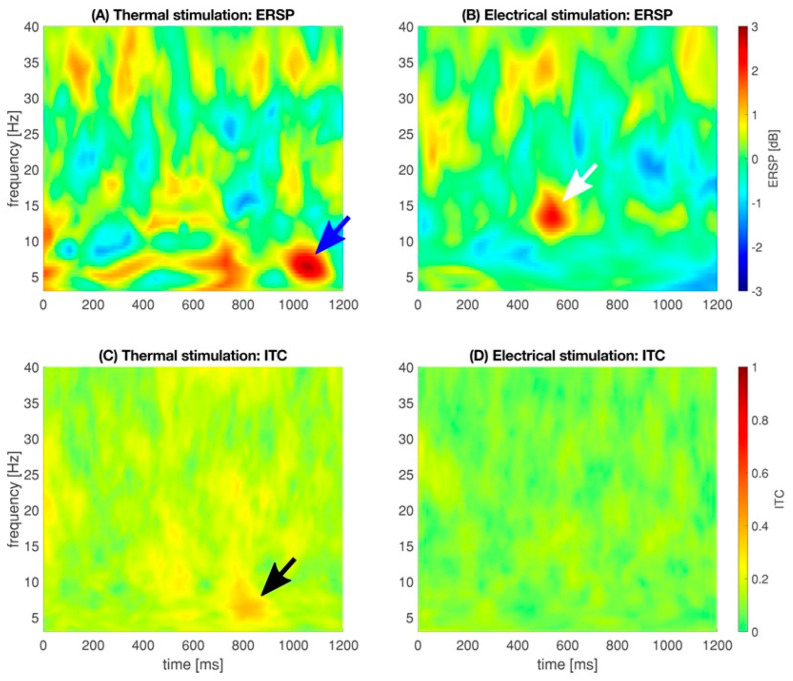
Median oscillatory responses as event-related spectral perturbation (ERSP, (**A**,**B**)), indicating the phase response, i.e., the oscillatory changes at a given time and frequency as a response to the stimulus. Inter-trial coherence (ITC, (**C**,**D**)), indicating the degree of phase-locking, i.e., the phase distribution of the stimulus across all trials. Blue arrow: local spectral maximum of 2.79 dB [−2.15 dB/5.16 dB] at 6.41 Hz and 1052 ms. White arrow: local spectral maximum of 2.23 dB [0.68 dB/2.23 dB] at 13.24 Hz and 543 ms. Black arrow: local ITC maximum of 0.38 [0.27/0.42] occurred at 6.41 Hz and 797 ms.

**Figure 5 animals-13-02839-f005:**
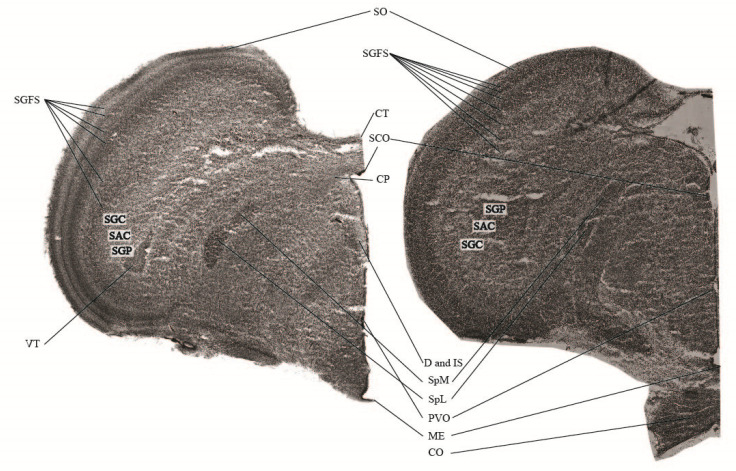
Anatomical differences of the embryonal brain: The two frontal sections (100 µm) from the central brain area represent the neuronal development of the embryonal brain from ED13 (**right**) and ED19 (**left**). Abbreviations: CO: Chiasma opticum, CP: Commissura posterior [caudalis] (Posterior commissure), CT: Commissura tectalis, D: Nucleus of Darkschewitsch; Nucleus paragrisealis centralis mesencephali (ICAAN), IS: Nucleus interstitialis (Cajal), ME: Eminentia mediana (Median eminence), PVO: Organum paraventriculare (Paraventricular organ), SAC: Stratum album centrale, SCE: Stratum cellulare externum, SCO: Organum subcommissurale (Subcommissural organ), SGC: Stratum granimalsiseum centrale, SGFS: Stratum griseum et fibrosum superficiale, SGP: Stratum griseum periventriculare, SO: Stratum opticum, SpL: Nucleus spiriformis lateralis, SpM: Nucleus spiriformis medialis, VT: Ventriculus tecti mesencephalic. The anatomical nomenclature was referred to anatomical atlases [15,46].

**Table 1 animals-13-02839-t001:** Summarized number of animals used for the different experimental approaches.

Stage (ED)	ED7	ED8	ED9	ED10	ED11	ED12	ED13	ED14	ED15	ED16	ED17	ED18	ED19	TOTAL
Animals total	36	9	8	9	18	56	57	14	13	17	18	40	66	361
Discarded	5	-	-	-	5	12	9	1	2	3	1	16	22	76
Evaluated	10	9	8	9	13	21	32	13	9	14	17	21	29	205 ^1^
Random EEG	-	-	-	-	-	-	-	-	-	-	-	2	13	15
Onset EEG	-	-	-	-	-	-	12	14	-	-	-	-	-	26
Electrical stimulation	8	6	4	5	9	6	15	6	4	7	11	12	10	103
Thermal stimulation	2	3	4	4	4	3	3	7	5	7	6	7	6	61
Histology	21	-	-	-	-	23	16	-	2	-	-	3	15	80

Discarded: From 361 embryos initially prepared, 76 were used for the establishment of the recording routines or discarded due to a low online signal/noise ratio. Evaluated: A total of 229 recording sessions deriving from ED07–ED19 were pre-analyzed for further analysis. Pre-analysis resulted in 15 animals for Random EEG (random hyperpallial EEG placements), 26 animals for Onset EEG (additional recording to evaluate ED12 and ED13), 103 animals for Electrical Stimulation, 61 animals for Thermal stimulation (^1^ 24 recordings from the initially evaluated animals were rejected after re-evaluation of the EEG quality). For histology, 80 embryos were used, partly originating from animals used for EEG recordings, partly especially prepared for histology.

**Table 2 animals-13-02839-t002:** Medians and percentiles [25%, 75%] of the PSD delta band (1–4 Hz) from the 6 randomly chosen embryos at each developmental stage, as shown in Figure 3.

STAGE (ED)	ED07	ED08	ED09	ED10	ED11	ED12	ED13	ED14	ED15	ED16	ED17	ED18	ED19
**MEDIAN DELTA POWER**	4.970	1.194	1.532	1.322	7.427	10.555	30.032	28.602	28.988	28.846	32.167	30.848	28.796
**[25%] PERCENTILE** **[75%] PERCENTILE**	3.231 5.972	0.338 1.968	0.181 2.141	0.788 3.377	4.582 10.873	6.274 12.567	28.906 30.656	26.767 28.842	26.640 30.442	28.003 30.497	28.551 32.530	30.564 31.861	27.526 33.304

## Data Availability

Raw data are available upon reasonable request to the corresponding author.

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
