# Peer review of "Nociception in Chicken Embryos, Part II: Embryonal Development of Electroencephalic Neuronal Activity In Ovo as a Prerequisite for Nociception"

_animals, 2023, doi:10.3390/ani13182839_

Round 1

Reviewer 1 Report

Following Part 1 of this study on finding the nociception onset during embryonic development in chicken, aiming to increase knowledge in this regard and thus improve animal welfare protection, it is noticeable that this study is far better developed than Part I. The methodology is clear, and methods/techniques were very well presented. Authors portrayed a sound and a scientifically enjoyable study.

Nonetheless, I propose a more adept presentation of the statistical analyses. There appears to be a resemblance to those performed in Part 1, where the analysis shown lacks rigor and does not align with the overall quality of this paper.

Author Response

Dear Reviewer, please see the attachment for our answers.

Please use Version 2 (updated).

Thank you very much for your precious support

Best regards

Thomas Fenzl

Reviewer 2 Report

The Authors in the presented work touch on an important topic - the ability to feel pain by chicken embryos. Chicken embryos are a biological research model, so a thorough study of their nervous system is very important.

As for the article itself, I have a few comments:

In which ED were thermal and electrical stimulation performed?

Was histological analysis performed for phases earlier than ED13?

Section 4.6 - I would consider moving it to the methods section (2.6)

Verse 94 - typo in the word Germany

Can you tell if any of these stimuli caused pain? Are there any known EEG patterns for chickens that indicate pain?Did the obtained EEG reading post-stimulation feel pain to the embryos? Are there reading thresholds that would definitely suggest discomfort?

The conclusions should be more specify.

Author Response

Dear Reviewer, please see attachment for our answers.

Please use Version 2 (updated).

Thank you very much for your precious support

Best regards

Thomas Fenzl
